# Analysis of the Microbial Diversity and Population Dynamics during the Pulque Fermentation Process

Fernando Astudillo-Melgar, Georgina Hernández-Chávez, María Elena Rodríguez-Alegría, Francisco Bolívar and Adelfo Escalante *

Departamento de Ingeniería Celular y Biocatálisis, Instituto de Biotecnología,
Universidad Nacional Autónoma de México, Cuernavaca 62210, Morelos, Mexico
* Correspondence: adelfo.escalante@ibt.unam.mx

**Abstract:** Pulque is the most-studied traditional Mexican alcoholic beverage prepared by fermentation of the fresh sap (aguamiel, AM) extracted from different *Agave* species (maguey) cultivated for pulque production. This beverage has been produced mainly in the Central Mexico Plateau since pre-Columbian times. In this contribution, we report the analysis of the bacterial and fungal diversity through 16S rRNA gene V3–V4 fragment amplicon and ITSR1 sequencing associated with the tissue of the walls (metzal) of the cavity or cajete, where the sap accumulates in producing plants for its daily extraction, in AM, and during four fermentation stages for pulque production. The results led to determining which microorganisms detected in the plant tissue are present in AM and maintained during the fermentation process. The results showed that eight bacterial OTUs (*Lactobacillus*, *Leuconostoc*, *Weisella*, *Lactococcus*, *Acetobacter*, *Gluconobacter*, *Zymomonas*, and *Obesumbacterium*) and five fungal OTUs (*Kazachstania*, *Kluyveromyces*, *Saccharomyces*, *Hanseniaspora*, and the OTU O_Saccharomycetales) were present from metzal to AM and during all the stages of the fermentation analyzed. The detected diversity was considered the microbial core for pulque fermentation, comprising up to ~84% of the total bacterial diversity and up to ~99.6% of the fungal diversity detected in the pulque produced from three plants of *A. salmiana* from the locality of Huitzilac, Morelos, Mexico. This study provides relevant information on the potential microorganisms responsible for pulque fermentation, demonstrating that the core of microorganisms is preserved throughout the elaboration process and their association with the AM and fermented beverage physiochemical profile.

**Keywords:** pulque; traditional fermented beverage; maguey; microbial core; massive sequencing; 16S rRNA V3–V4; ITSR1

## 1. Introduction

Pulque (PQ) is a traditional Mexican alcoholic beverage produced from the fermentation of fresh sap (aguamiel, AM) extracted from different species of *Agave* plants (maguey), mainly *A. atrovirens*, *A. salmiana*, *A. mapisaga*, and *A. americana* cultivated for pulque production [1–4]. This beverage is produced mainly in the Central Mexican Plateau [3], and its current traditional production process remains practically identical to its production in pre-Columbian times. Archeological evidence has demonstrated its production as an extended practice since 500 BC–100 AD [1,2,5]. The chemical identification of microbial cell wall hopanoids in PQ production/consumption pottery dated to 200–550 AD, at the height of the power and dominion of the ancient Teotihuacan culture, suggest its production and consumption as a relevant agricultural activity in the Teotihuacan Valley (Central Mexican Highlands), a zone with low rainfall and limited groundwater [6]. Associated with its ancient origin, PQ is possibly the most-studied traditional fermented beverage in Mexico from different points of view: cultural, religious, nutritional, traditional pharmacopeia, economics, and microbiological [2,7,8].

Briefly, the PQ fermentation process starts with the collection and the spontaneous fermentation of high-quality fresh AM to produce a seed, which is transferred into

a fermentation vat, into which fresh collected AM (at daybreak and sunset) is poured and fermented until the beverage attains its distinctive sensorial characteristics: production of ethanol and $CO_2$, slightly acidic, viscous, and its characteristic bouquet [1,2,9]. PQ fermentation is performed by a complex microbial diversity composed of bacteria and yeasts proposed as naturally associated with the sap, by microorganisms incorporated in the seed, and probably from the plant. This complex microbiome catabolizes the available sugars in the AM, mainly sucrose, glucose, and fructose, to yield 4–7% ethanol produced mainly by *Zymomonas mobilis* and *Saccharomyces* species. The slightly acid characteristic of the beverage (pH ranging from 3.5 to 4.2) results from the production of lactic and acetic acids produced by several lactic bacteria (mainly *Lactobacillus* sp., *Lactococcus* sp., and *Leuconostoc* sp.) and acetic acid bacteria (*Acetobacter* sp., *Acinetobacter* sp., and *Gluconobacter*), and finally, the characteristic viscosity results from the production of dextran (from the glucose-moiety from sucrose) by several species of *Leuconostoc* and levan (from the fructose-moiety from sucrose) by some *Leuconostoc* species and *Z. mobilis* [10–13].

Recent shotgun metagenomic sequencing in five different stages of PQ fermentation resulted in the taxonomic profiling and determination of the relative abundance of the microbial community present in fresh collected AM and during fermentation (T0, T3, T6, and fermented PQ) from the locality of Huitzilac, Morelos state, Mexico. The results identified six genera: *Acinetobacter*, *Zymomonas*, *Lactobacillus*, *Lactococcus*, *Leuconostoc*, and *Saccharomyces*, and ten species: *Acinetobacter boissieri*, *A. nectaris*, *Lactobacillus sanfranciscensis*, *Lactococcus plantarum*, *L. citreum*, *L. piscium*, *L gelidum*, *Z. mobilis*, and the yeast *S. cerevisiae*, all present $\geq$ 1% in at least one analyzed stage [11]. Changes in the relative abundance of specific genera and species correlated with the sugar content in AM (sucrose, glucose, and fructose) and its consumption during fermentation, as in ethanol, lactic, and acetic acid production [11].

The maguey plant used for AM production provides not only the sugars available as the carbon source for the pulque fermentation process (sucrose, glucose, and fructose) but also various fructooligosaccharides (FOS) resulting from the hydrolysis of the inulin of the plant, degraded during the fermentation process [13,14]. However, there are no reports on the possible contribution of microorganisms naturally associated with the plant to the microbial diversity present in AM during its accumulation in the plant or their possible participation in pulque fermentation. A previous report on the analysis of the microbiota associated with the plant (root and leaf) *A. salmiana* reported 33 distinct prokaryotic phyla associated with the plant, but just the Proteobacteria, Actinobacteria, Firmicutes, and Bacteroidetes comprised > 80% of the associated prokaryotic diversity. The endophytic bacterium *Leuconostoc* was also isolated from *A. tequilana* [15]. These results suggest that the plant could become the source of some bacterial and fungal species for pulque fermentation. In this contribution, we report the bacterial and fungal diversity throughout the complete process of pulque fermentation, including, for the first time, the analysis of the microbial diversity associated with the tissue of the walls (metzal) of the cavity (cajete) where AM accumulates in producing plants, its fresh collected AM, and in four stages of the fermentation, T0, T3, T6 h, and overnight fermented PQ (~12 h), in a controlled fermentation in the laboratory through a 16S rRNA gene V3–V4 fragment amplicon and ITSR1 sequencing to determine the core of microorganisms responsible for the fermentation of pulque, including those coming from the plant, and its presence in AM and during the fermentation process.

## 2. Materials and Methods

### 2.1. Sample Collection

Samples of overnight accumulated aguamiel were collected from the cavity (cajete) of three selected plants (Figure 1D) at daybreak using the traditional sap extraction tool until the cavity was empty. The sap was transferred immediately into sterile plastic bags and placed in ice in a cooler. Once the overnight accumulated sap was extracted from each plant, the vegetal tissue of the walls of the cavity (metzal) (Figure 1D,E) was extracted by

the traditional procedure using a scraping tool, placed in sterile plastic bags, and stored in ice as above. Overnight fermented pulque (~12–14 h) samples were withdrawn directly from the fermentation container by the producer, placed in sterile bags, stored in ice as above, and transported to the laboratory (approximately 25 min by car) [10,11]. Samples were provided by Mr. Salvador Cueto, a local traditional pulque producer from Huitzilac (north 19°01′49.3″ west 99°15′53.1″, 2561 MASL), a town 10 km north of the Morelos state capital, Cuernavaca. Samples were collected from three different maguey plants *Agave salmiana* 5–6 years old. These plants were just at the start of their producing lifetime, they were in the same maguey plantation, each one was sampled over three different dates in the fall season (Figure 1, Table 1).

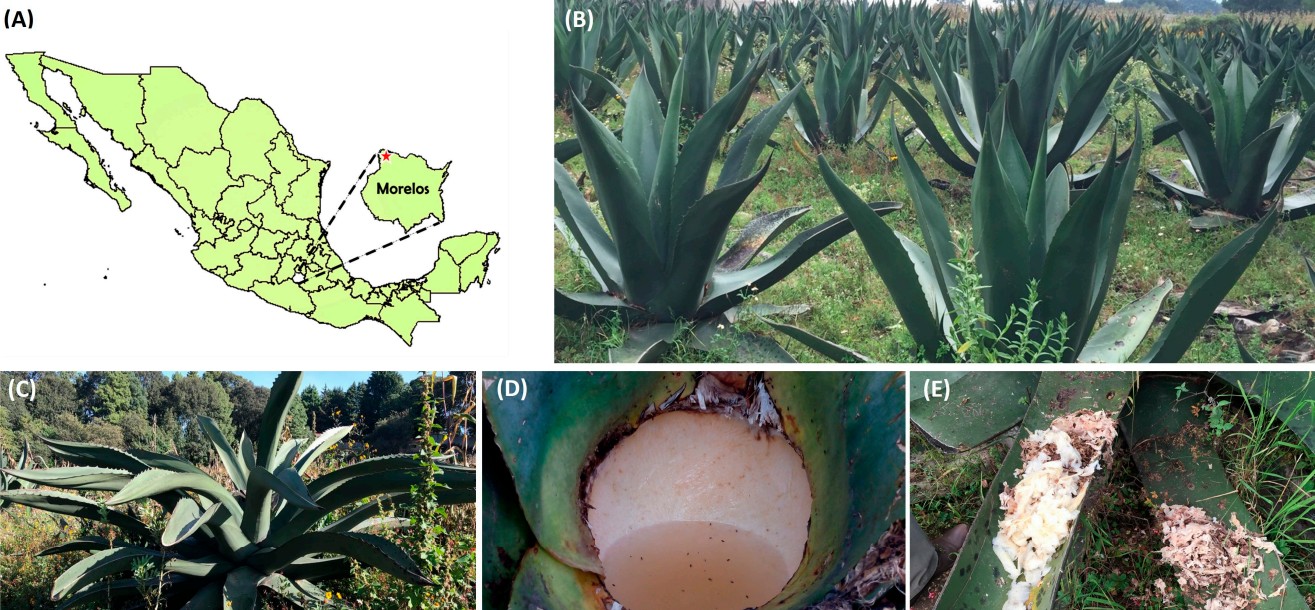

**Figure 1.** Sampling location. (**A**) Huitzilac, Morelos (red star). (**B**) Maguey cultivation zone. (**C**) One of the three plants sampled for aguamiel and metzal. (**D**) Cajete or cavity where aguamiel accumulates after scraping the walls, (**E**) Sample of the scraped metzal tissue.

**Table 1.** Sample collection characteristics.

| Parameter | Sample 1 | Sample 2 | Sample 3 |
|---|---|---|---|
| Date | 7 November 2019 | 11 November 2019 | 14 November 2019 |
| Environmental temperature | 22 °C | 21 °C | 20 °C |
| Environmental humidity | 87% | 89% | 88% |
| Ambient pressure | 732 atm | 731 atm | 731 atm |
| Weather | Foggy | Sunny | Cloudy |

Data acquired from MeteoRed.mx, accessed on 14 November 2019.

In a sterile environment in a laminar flow hood and using sterile materials, 50 g of metzal tissue from each plant was processed (squeezed) with a metal lemon squeezer. The resultant extract was differentially centrifuged [16] at 1000 rpm for 15 min at 4 °C to separate vegetal debris. The resultant pellet containing plant tissue debris was discarded, and the supernatant (containing microbial cells) was centrifuged again at 10,000 rpm for 20 min at 4 °C. The pellet was dry-ice-frozen and maintained at −70 °C until further total DNA extraction (Section 2.6), whereas the supernatant was frozen until its analysis through HPLC for metabolite content analysis [10,11].

### 2.2. Pulque Fermentation in the Laboratory

Controlled laboratory fermentations were performed as previously [10,11] for each AM sample collected. Briefly, the fermentation was carried out in a sterile plastic container by mixing AM and fermented PQ with a proportion of 2:3 (AM: PQ) (total volume ranging from 3 to 5 L depending on the volume of AM collected, which varied between the three sampled plants) for 6 h without stirring at room temperature. Aliquots of 10 mL (by triplicate) were withdrawn at 0, 3, and 6 h of fermentation and centrifuged at 10,000 rpm for 20 min at 4 °C. Pellets and supernatants were separated and frozen at −20 °C for further analysis. Samples of AM and PQ were also processed as described.

### 2.3. Physicochemical Parameters Analysis (pH, Temperature, $CO_2$, Dissolved Oxygen), Ashes, and Biomass

Temperature and pH were monitored in AM and PQ after inoculation and 0, 3, and 6 h of fermentation as described previously [10,11]. Biomass and ashes were determined using the methodology reported in the Mexican Official Norm (NMX-V-017-1970, https://dof.gob.mx/nota_detalle.php?codigo=5594808&fecha=11/06/2020#gsc.tab=0 (accessed on 7 April 2022) and AOAC (Supplementary File S1). The concentrations of atmospheric $CO_2$ and dissolved oxygen (DO) were measured using an Applikon reactor (myControl for MiniBio Reactors) of 250 mL with $CO_2$ and DO sensors.

### 2.4. HPLC Quantification: Sucrose, Glucose, Fructose, Glycerol, Organic Acids, and Ethanol

Sugars, organic acids, and ethanol concentrations from the AM, PQ, metzal, T0, T3, and T6 stages were determined for all samples in a Waters HPLC system equipped with an Aminex column for fermentation analysis. A total of 1 mL of each sample was centrifuged at $10,000 \times g/5$ min; supernatants were filtered through a 0.45 μm membrane, and a 20 μL aliquot was injected into the column. Glucose, fructose, sucrose, and glycerol were quantified using an Aminex HPX-87P (BioRad, Hercules, CA. USA) column with an IR detector. Acetate, lactate, succinate, and ethanol concentrations were quantified using an Aminex HPX-87H (Biorad) column and a $UV_{210nm}$ detector [10,11].

### 2.5. Fructooligosaccharides Identification

All samples' FOS were identified using IC Dionex™ with a Carbopack PA200 column (2 mm diameter × 250 mm length). The mobile phases used were Solution 1 NaOH 0.09M, $C_2H_3NaO_2$ 0.5 M, and Solution 2 NaOH 0.09 M with a 1–20% gradient of Solution 1. Orafti P95 5 mg/mL was used as standard [13].

### 2.6. 16S rRNA Gene V3–V4 Fragment Amplicon and ITSR1 Library Preparation and Sequencing

The DNA was extracted from 10 mL of AM, T0, T3, T6, and fermented PQ stages using the Power Food microbial DNA isolation Kit (MoBio, Carlsbad, CA. USA) according to the manufacturer's protocol. Total DNA from metzal was extracted from 5 mL of the extract obtained as described above (Section 2.1). The extracted DNA was analyzed using 1% agarose gel electrophoresis and quantified using a Nanodrop 2000c (Thermo Fisher Scientific, Waltham, MA. USA). The samples were sequenced on an Illumina MiSeq Sequencer (Instituto de Biotecnología, Universidad Nacional Autónoma de México), producing paired-end 250 bp reads. 16S rRNA gene V3–V4 fragment amplicon and ITSR1 amplicon libraries were prepared following Illumina protocols (https://support.illumina.com/documents/documentation/chemistry_documentation/16s/16s-metagenomic-library-prep-guide-15044223-b.pdf (accessed on 7 April 2022)). Three technical replicates were sequenced at each stage (AM, metzal, T0, T3, T6, and overnight fermented PQ). The 16S rRNA gene V3–V4 fragment amplicons were sequenced for bacteria using the V3 and V4 region primers F (5′-TCGTCGGCAGCGTCAGATGTGTATAAGAGACAGCCTACGGGNGGCWGCAG-3′) and R (5′-GTCTCGTGGGCTCGGAGATGTGTATAAGAGACAGGACTACHVGGGTATCTAATCC-3′). The fungal internal transcribed spacer (ITSR1) was amplified using a mixture of primers eight F: 1_CTTGGTCATTTAGAGGAAGTAA; 2_CTCGGTCATTTAGAGGAAGTAA;

3_CTTGGTCATTTAGAGGAACTAA; 4_CCCGGTCATTTAGAGGAAGTAA; 5_CTAGG
CTATTTAGAGGAAGTAA; 6_CTTAGTTATTTAGAGGAAGTAA; 7_CTACGTCATTTA
GAGGAAGTAA; 8_CTTGGTCATTTAGAGGTCGTAA and seven R: 1_GCTGCGTTCTTCA
TCGATGC; 2_GCTGCGTTCTTCATCGATGG; 3_GCTACGTTCTTCATCGATGC; 4_GCT
GCGTTCTTCATCGATGT; 5_ACTGTGTTCTTCATCGATGT; 6_GCTGCGTTCTTCATCGTT
GC; 7_GCGTTCTTCATCGATGC [17].

### 2.7. Bioinformatics and Statistical Analysis

For 16S rRNA gene V3–V4 fragment amplicon and ITSR1 sequence analysis, we used a previously reported pipeline as follows [17–20]: raw sequencing data quality was improved as described previously [11,17], and quality-improved read sets were inspected via FASTQC (https://www.bioinformatics.babraham.ac.uk/projects/fastqc/ (accessed on 5 September 2022)). Identified chimeric sequences were cleaned from the dataset through blast fragments and ChimeraSlayer with the QIIME's parallel_identify_chimeric_seqs.py script [18,20].

The sequences were analyzed using QIIME (Quantitative Insights Into Microbial Ecology) version-1.9.1 software in Python 2.7 [18,20]. The sequences were clustered into operational taxonomic units (OTUs), and the marge was performed for the SeqPrep method. Bacterial identification was performed using the SILVA database (SILVA_132) with a reference with a range of 97% similarity and the closed system with the command pick_closed_reference_otus.py. For fungi, taxonomy assignment was performed using UNITE 2020 [19]. For all datasets, the data filtering option of 0.01% was used in abundance because it is reported that filtering a database decreases the estimation error [21,22].

Alpha diversity was evaluated using alpha diversity metrics such as the Shannon–Wiener index, Simpson index, OTUs_observed, and Chao1 value. The result of each metric was analyzed through ANOVA, applying the Tukey–Kramer test (0.95 confidence interval) to estimate the significant difference between the samples. Beta diversity was calculated using the Bray–Curtis matrix and generating principal coordinates analysis (PCoA) plots from QIIME [17].

## 3. Results

### 3.1. Physical Parameters of Sampled AM, PQ, and Laboratory Fermentation

Three fermentations for PQ production were performed as described previously [10,11] immediately after each sample of AM and overnight fermented PQ arrived at the laboratory. Collected AM showed its typical slightly cloudy color with a sweet herbal aroma. Overnight fermented PQ showed its characteristic milky white, foamy appearance, slight acidity, and alcoholic content. The analyzed laboratory fermentation stages were T0 (AM inoculated with previously fermented PQ), T3, T6 (three and six hours of fermentation, respectively), metzal, AM, and overnight fermented PQ.

The three laboratory fermentations showed similar pH, % DO, and % $CO_2$ values. AM showed an initial pH = 4.5, which, after inoculation with overnight fermented PQ (T0) (pH = 3.8), decreased to 4.1 due to the addition of fermentation products in PQ. The pH of the T6 stage was 4.0. The % DO during the laboratory fermentation decreased from 3.5% (T0) to 0.8% at the T6 stage. The % $CO_2$ decreased from 0.03% (T0) to 0.019% (T6) (Figure 2A). The AM sample had an initial temperature of 20 °C, which, after inoculation, increased to 24 °C and then gradually up to 25 °C after 3 h and 26 °C after 6 h. In contrast, the temperature of the fermented PQ was 20 °C, indicating relevant microbiological activity during fermentation (Figure 2A).

Regarding total solids, biomass, and ashes, AM showed an average of 124.5 g/L of total solids, 121.81 g/L of biomass, and an ashes content of 277.7 mg/100 mL. After 6 h of fermentation, total solids and biomass decreased to 43.5 g/L and 41.3 g/L, respectively, and ashes showed a final value of 221.9 mg/100 mL (Figure 2B). Overnight fermented PQ showed 36.9 g/L and 34.73 g/L of total solids and biomass, respectively, whereas the ashes content was 219.7 mg/100 mL.

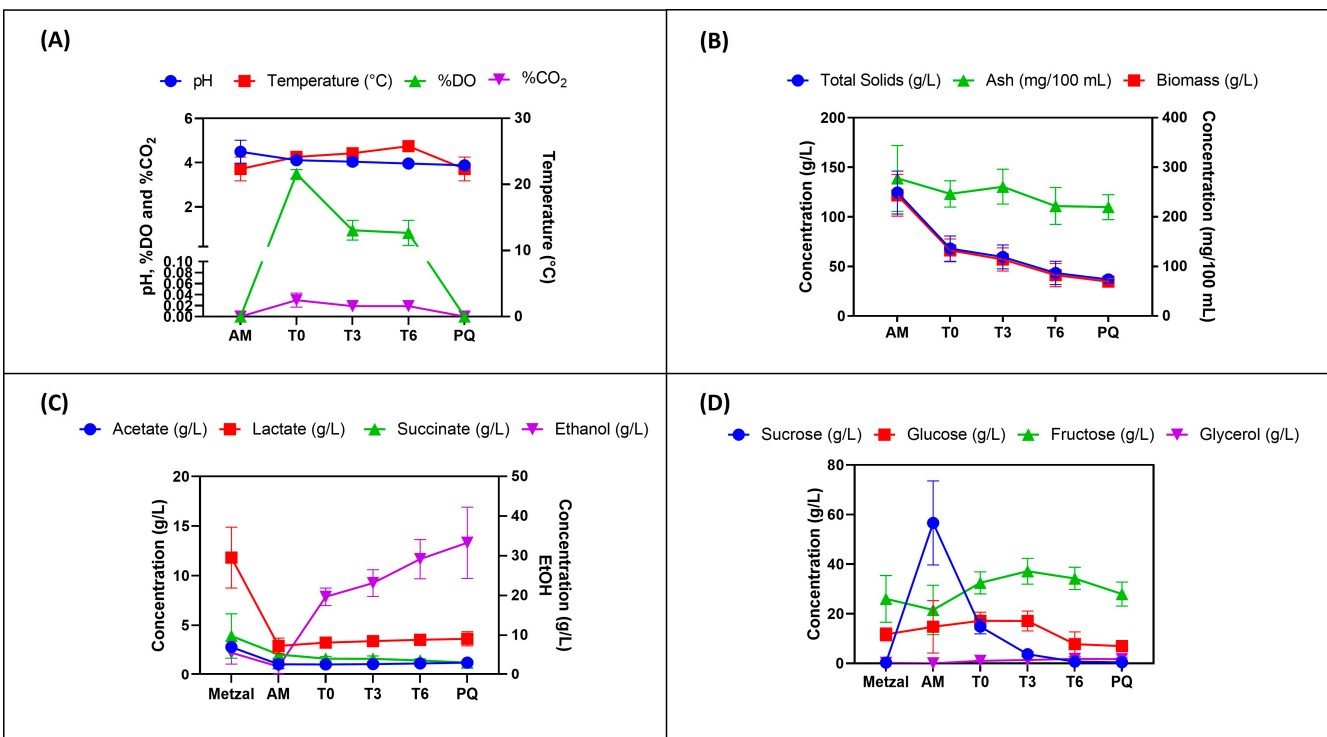

**Figure 2.** Metabolic characteristics and physical parameters of metzal, AM, and PQ fermentation. (**A**) Physical parameters. (**B**) Biomass, total solids, and ash content. (**C**) Organic acids and ethanol. (**D**) Carbohydrates. The graphs represent the average of the triplicates analyzed.

The content of lactic acid, ethanol, acetic acid, and succinic acid as the main fermentation products were determined in metzal, AM, during the laboratory fermentation stages (T0–T6), and in overnight fermented PQ. Lactic acid was the most abundant fermentation product in metzal, showing an average of 11.82 g/L. In AM, there was 2.92 g/L lactic acid, and its content increased during the fermentation to 3.5 g/L in T6, whereas overnight fermented PQ had 3.54 g/L (Figure 2C). Ethanol was the second most abundant product in metzal with a concentration of 5.60 g/L and 2.1 g/L in fresh collected AM, which increased during fermentation from 19.7 g/L (T0) to 29.2 g/L (T6), whereas overnight fermented pulque showed 33.3 g/L. Succinic acid was detected in quantities of 3.9 g/L in metzal and 2.06 g/L in AM. Its content essentially did not change during the laboratory fermentation stages from fermented pulque. Finally, acetic acid was detected in a higher concentration in metzal, whereas AM, the laboratory fermentation stages, and fermented PQ, showed similar concentration values (Figure 2C).

Concerning the average concentration of analyzed sugars, sucrose was the most abundant in AM but was practically absent in the metzal (0.30 g/L). During laboratory fermentation, the content of this sugar decreased from 14.8 g/L in T0 to 0.69 g/L in T6 and 0.43 g/L in the overnight fermented PQ. The hydrolysis of sucrose during pulque fermentation resulted in glucose and fructose moieties (Figure 2D). Glucose was detected in quantities of 11.7 g/L in metzal and 14.8 g/L in AM. Its content in T0 was 17.16 g/L, which remained constant in T3 (17.10 g/L) but decreased to 7.78 g/L in T6 and 6.99 g/L in overnight fermented PQ (Figure 2D). Fructose was detected in quantities of 26 g/L in metzal and 21.55 g/L in AM. During the laboratory fermentation, fructose was detected in quantities of 32.48 g/L in T0. Its concentration increased to 34.23 g/L at the end of fermentation (T6) and 27.93 g/L in the overnight fermented pulque. Glycerol, likely produced by yeasts, was detected in quantities of 0.21 g/L in metzal and was absent in AM, but its concentration increased during laboratory fermentation from 1.06 g/L to 1.81 g/L at T6 and remained at 1.72 g/L in the overnight fermented PQ (Figure 2D).

The FOS were identified in metzal, AM, laboratory fermentation stages T0, T3, T6, and in overnight fermented PQ through HPAEC-PAD. The peaks of those FOS detected in all the samples were labeled from 1 to 23. Among them, four peaks were identified, the nystose (peak 5), fructosyl-nystose (peak 8), ketohexose (peak 12), and ketoheptose (peak 16), respectively (Figure 3). Metzal showed 21 of the total detected FOS, including all the identified ones. Unidentified FOS in metzal were the peaks 1–4, 6, 7, 9–11 (FOS with less than five fructose moieties) and peaks 12–15 and 17–21 (FOS with 5–7 fructose moieties). Nystose was the only FOS detected in all the analyzed samples. Unidentified peaks 1–4 and 7 were present in all samples; peak 6 was only detected in metzal and AM. Peaks 9, 13, and 17 were identified in metzal, AM, and T0–T6 samples but not in overnight fermented pulque. Interestingly, peak 22 was detected in AM and fermentation stages T0–T6, whereas peak 23 was detected in pulque samples (Figure 3). Supplementary Figure S1 shows the complete chromatograms obtained through HPAEC-PAD corresponding to all analyzed triplicate samples.

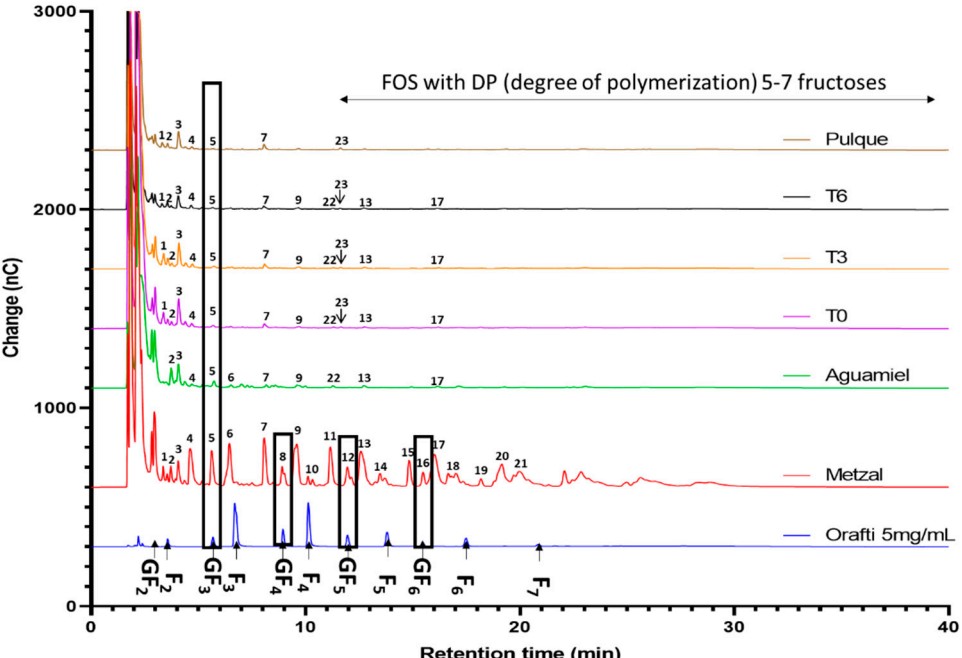

**Figure 3.** Fructooligosaccharide identification through HPAEC-PAD in metzal, AM, and fermentation stages. The black rectangles show the identified FOS in the analyzed samples. The chromatograms' peak numbering (1–23) shows all the identified FOS. The blue line corresponds to FOS standard (Orafti): $GF_2$: 1-Ketose, $F_2$: Inulobiose, $GF_3$: Nystose, $F_3$: Inulotriose, $GF_4$: Fructosyl-nystose, $F_4$: Inulotetrose, $GF_5$: Ketohexose, $F_5$: Inulopentose, $GF_6$: Ketoheptose, $F_6$: Inulohexose, and $F_7$: Inuloheptose. The chromatogram for each sample is the average of all the individual analyses, e.g., metzal = average of runs metzal 1, metzal 2, and metzal 3. Supplementary Figure S1 shows the complete chromatograms obtained through HPAEC-PAD corresponding to all analyzed triplicate samples.

### 3.2. Total DNA Extraction and Sequencing and Taxonomy Assignment

For the first time in the study of the PQ microbiome, the total DNA was extracted and sequenced in triplicate from samples of metzal, aguamiel, the stages of each fermentation developed in the laboratory, T0, T3, and T6, and in the overnight fermented PQ, resulting in a total of 18 sequenced samples for each plant sampled. Each sample of extracted DNA was used for library preparation and sequencing of the 16S rRNA gene V3–V4 fragment amplicon regions for bacteria and ITSR1 for fungi using the Illumina MiSeq platform. We assessed the taxonomic profile and relative abundance of bacteria and fungi in metzal, AM, and all pulque stages using the SILVA_132 database for bacterial taxonomic classification and UNITE 2020 for fungi, allowing the identification at the genus level

and, in some instances, at the species level. The raw sequence data of each sequenced sample are available in the Portal of the National Center for Biotechnology Information, BioProject PRJNA921737.

### 3.3. Taxonomy Assignment and Diversity Analysis

Raw data from a total of 1,098,819 bacterial sequences were used from all the analyzed samples, allowing the identification of 46 unique OTUs assigned by debugging the database to 0.01% to eliminate chimeras and sequencing errors [20]. For bacteria, at the genus level, the following eight genera were detected in all the analyzed samples (metzal, AM, fermentation stages T0, T3, T6, and overnight fermented PQ) (Figure 4): *Zymomonas* (58.76% of the total bacterial diversity), *Lactococcus* (8.04%), *Weissella* (7.08%), *Leuconostoc* (3.91%), *Lactobacillus* (0.05%), *Acetobacter* (5.4%), *Gluconobacter* (0.06%), and *Obesumbacterium* (0.38%), with these considered as the bacterial core for pulque fermentation. Those microorganisms not detected in all the analyzed samples were excluded from the bacterial core for pulque and included as members of the Others group. The complete identity assigned to these OTUs and their relative abundance in metzal, AM, and during the fermentation stages is shown in Supplementary Table S1.

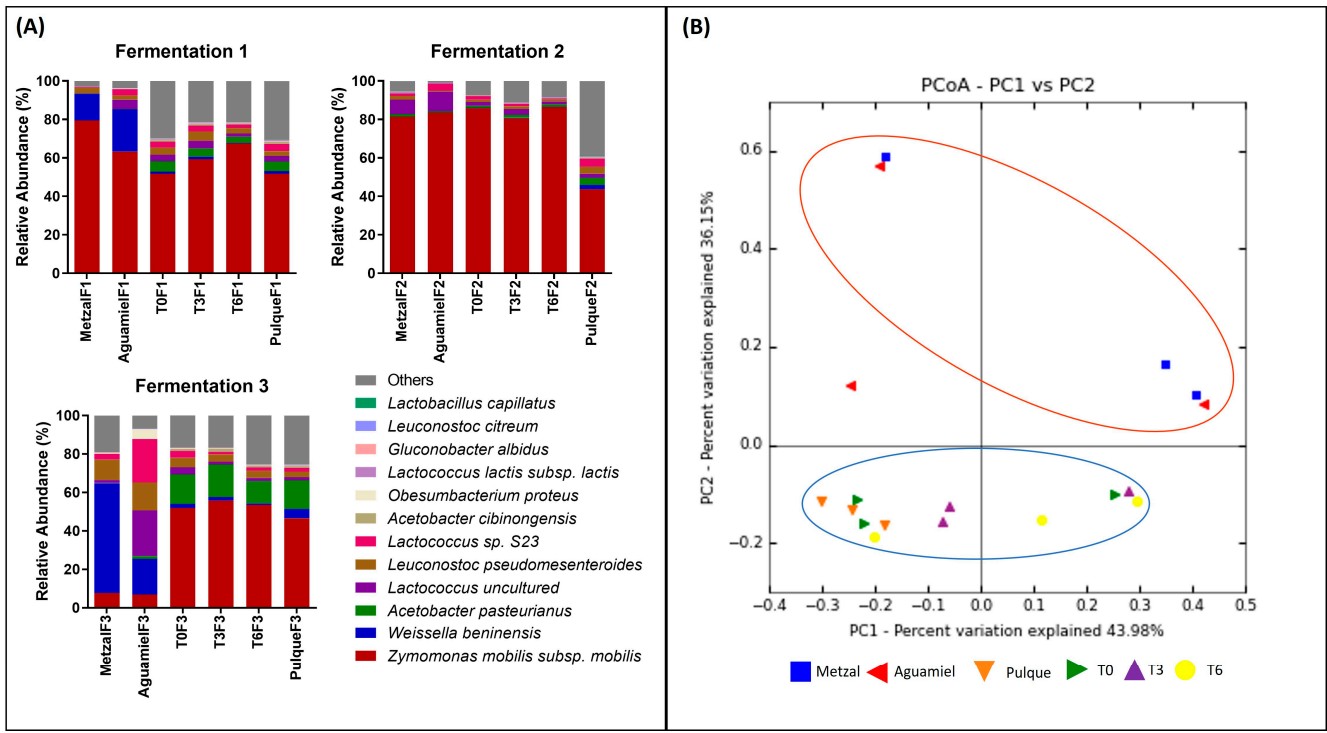

**Figure 4.** Bacteria diversity during pulque fermentation. (**A**) The relative abundance profile of Bacteria is shown for each of the three fermentations using the SILVA132 database. (**B**) Bacteria PCoA plots based on the Bray–Curtis analysis. The red circle shows the association between AM and metzal samples, whereas the blue circle shows the association between the fermentation stages. The graphs represent the microorganisms with more than 0.01% of relative abundance. The complete relationship of those microorganisms included in the group named Others is shown in the Supplementary Files (Table S1).

Based on calculated diversity indexes for bacteria (Table 2), overnight fermented PQ showed the highest diversity, followed by T6, T3, T0, metzal, and AM. For overnight fermented PQ Observed_otus$_{PQ}$ = 96.67, Chao1$_{PQ}$ = 156.46, H$_{PQ}$ 0 3.06, and D$_{PQ}$ = 0.75. At the genus level, the proposed bacterial core in fermented PQ was composed of *Zymomonas* (47.34%), *Acetobacter* (8.70%), *Weissella* (2.82%), *Leuconostoc* (3.07%), *Obesumbaterium* (0.14%), *Gluconobacter* (0.09%), and *Lactobacillus* (0.1%). In T6, Observed_otus$_{T6}$ = 86.33, Chao1$_{T6}$ = 161.53, H$_{T6}$ = 1.97, and D$_{T6}$ = 0.49. At the genus level, the bacterial core was

composed of *Zymomonas* (69.25%), *Acetobacter* (5.91%), *Leuconostoc* (2.69%), *Lactococcus* (3.19%), *Weissella* (0.35%), *Gluconobacter* (0.05%), *Obesumbacterium* (0.05%), and *Lactobacillus* (0.02%). In T3, Observed_otus$_{T3}$ = 99.67, Chao1$_{T3}$ = 146.29, H$_{T3}$ = 2.24, and D$_{T3}$ = 0.55. For this fermentation stage, the bacterial core at the genus level was composed of *Zymomonas* (65.23%), *Acetobacter* (8.50%), *Leuconostoc* (3.42%), *Lactococcus* (5.05%), *Weissella* (1.10%), *Gluconobacter* (0.10%), *Obesumbaterium* (0.09%), and *Lactobacillus* (0.05%). For T0, the diversity indexes were Observed_otus$_{T0}$ = 80.33, Chao1$_{T0}$ = 128.17, H$_{T0}$ = 2.25, and D$_{T3}$ = 0.56. The bacterial core was composed of *Zymomonas* (63.18%), *Acetobacter* (7.97%), *Lactococcus* (6.14%), *Leuconostoc* (3.30%), *Weissella* (1.12%), *Obesumbacterium* (0.11%), *Gluconobacter* (0.07%), and *Lactobacillus* (0.04%). For metzal, Observed_otus$_{MZ}$ = 77, Chao1$_{MZ}$ = 126.22, H$_{PQ}$ = 1.56, and D$_{MZ}$ = 0.44. The bacterial core in metzal was composed of *Zymomonas* (56.28%), *Weissella* (23.56%), *Leuconostoc* (5.16%), *Lactococcus* (4.86%), *Acetobacter* (0.54%), *Obesumbacterium* (0.24%), *Lactobacillus* (0.05%), and *Gluconobacter* (0.05%). Finally, for AM, Observed_otus$_{AM}$ = 60.67, Chao1$_{AM}$ = 106.37, H$_{AM}$ = 1.96, and D$_{AM}$ = 0.56. The bacterial core in AM was composed of *Zymomonas* (51.27%), *Weissella* (13.54%), *Lactococcus* (22.99%), *Leuconostoc* (5.83%), *Obesumbaterium* (1.67%), *Acetobacter* (0.77%), *Lactobacillus* (0.03%), and *Gluconobacter* (0.01%). The individual values (% relative abundance) of bacterial diversity are in Supplementary Table S3. At the genus level, the eight proposed components of the bacterial core from the plant (metzal and AM) to the fermented beverage (stages T0, T3, T6, and overnight PQ) represent 83.68% of the total relative abundance detected (Figure 4, Table 3).

**Table 2.** Alpha diversity values.

| Diversity Index/Sample | Observed_OTUs | Chao1 | Shannon–Wiener | Simpson |
| --- | --- | --- | --- | --- |
| | | Bacteria | | |
| Metzal *n* = 3 | 77.0 ± 20.66 | 126.22 ± 31.11 | 1.56 ± 0.61 | 0.44 ± 0.17 |
| Aguamiel *n* = 3 | 60.67 ± 17.93 | 106.37 ± 38.66 | 1.96 ± 1.06 | 0.56 ± 0.28 |
| T0 *n* = 3 | 80.33 ± 9.71 | 128.17 ± 23.22 | 2.25 ± 0.96 | 0.56 ± 0.25 |
| T3 *n* = 3 | 99.67 ± 18.48 | 146.29 ± 43.47 | 2.24 ± 0.62 | 0.55 ± 0.17 |
| T6 *n* = 3 | 86.33 ± 15.50 | 161.53 ± 63.55 | 1.97 ± 0.77 | 0.49 ± 0.22 |
| Pulque *n* = 3 | 96.67 ± 22.81 | 156.46 ± 60.47 | 3.06 ± 0.18 | 0.75 ± 0.03 |
| | | Fungi | | |
| Metzal *n* = 3 | 15.66 ± 3.78 | 16.77 ± 5.39 | 1.09 ± 0.15 | 0.442 ± 0.04 * |
| Aguamiel *n* = 3 | 20.66 ± 3.51 | 21.66 ± 2.08 ** | 1.26 ± 0.55 | 0.44 ± 0.20 |
| T0 *n* = 3 | 12.33 ± 0.57 | 12.33 ± 0.57 | 0.68 ± 0.03 | 0.20 ± 0.008 |
| T3 *n* = 3 | 13.00 ± 0.00 | 13.00 ± 0.00 | 0.65 ± 0.03 | 0.19 ± 0.01 |
| T6 *n* = 3 | 13.00 ± 1.00 | 13.50 ± 1.80 | 0.61 ± 0.02 * | 0.18 ± 0.002 * |
| Pulque *n* = 3 | 13.33 ± 1.15 | 13.33 ± 1.15 | 0.77 ± 0.16 | 0.24 ± 0.06 |

An ANOVA statistical test with a 95% confidence interval and a Tukey test were used to obtain the values shown in this table. * Shows significant difference with $p < 0.01$. ** Shows significant difference with $p < 0.001$.

A total of 3,107,428 raw reads were used for fungi, and 134 OTUs were assigned at the species level. Filtering the abundance of 0.01% [21] resulted in the determination of 18 representative OTUs. Five genera, *Kluyveromyces marxianus*, *Saccharomyces cerevisiae*, *Kazachstania gamospora*, *Hanseniaspora* sp., and the OTU O_Saccharomycetales were detected in all the analyzed samples (metzal, AM, fermentation stages T0, T3, T6, and overnight fermented PQ) (Figure 5), and they were considered as the fugal core for pulque fermentation. As for bacteria, those microorganisms not detected in all the analyzed samples were included as members of the Others group and were not considered part of the fungal core of pulque. The complete identity assigned to these OTUs and their relative abundance is shown in Supplementary Table S1.

**Table 3.** The proposed microbial core for pulque fermentation present in all the analyzed samples.

|  | Core | Relative Abundance (%) |
|---|---|---|
| | *Zymomonas* | 58.76 |
| | *Lactococcus* | 8.04 |
| | *Weissella* | 7.08 |
| Bacteria | *Acetobacter* | 5.4 |
| (Genus level) | *Leuconostoc* | 3.91 |
| | *Obesumbacterium* | 0.38 |
| | *Gluconobacter* | 0.06 |
| | *Lactobacillus* | 0.05 |
| | Total | 83.68 |
| | *Saccharomyces cerevisiae* | 70.07 |
| | *Kluyveromyces marxianus* | 18.13 |
| Fungi | *Kazachstania gamospora* | 10.84 |
| | *Hanseniaspora* sp. | 0.27 |
| | O_Saccharomycetales | 0.27 |
| | Total | 99.58 |

The diversity indexes calculated for fungi indicated that AM showed the highest diversity, followed by metzal. During the fermentation stages, the fungal diversity remained practically constant (Table 2). In AM, Observed_otus$_{AM}$ = 20.66, Chao1$_{AM}$ = 21.66, H$_{AM}$ = 1.26, and D$_{AM}$ = 0.44. The fungal core was composed of *K. marxianus* (61.03%), *S. cerevisiae* (22.62%), *K. gamospora* (14.21%), *Hanseniaspora* sp. (0.45%), and O_Saccharomycetales (0.13%). In metzal, Observed_otus$_{MZ}$ = 15.66, Chao1$_{MZ}$ = 16.77, H$_{MZ}$ = 1.09, and D$_{MZ}$ = 0.44, and the fungal core was composed of *K. marxianus* (47.67%), *K. gamospora* (40.75%), *S. cerevisiae* (8.95%), O_Saccharomycetales (1%), and *Hanseniaspora* sp. (0.71%). In T0, Observed_otus$_{T0}$ = 12.33, Chao1$_{T0}$ = 12.33, H$_{T0}$ = 0.68, and D$_{T0}$ = 0.20. Remarkably, during fermentation (T0 to T6 and PQ stages), *S. cerevisiae* increased until it reached the maximum abundance for all observed fungi (>97.3%), and *K. gamospora* decreased from 14.21% in AM to 2.51% during fermentation and 4.37% by the end of fermentation (overnight fermented PQ) (Table 2) (Figure 6). For fungi, the five microorganisms proposed as the core from the plant (metzal and AM) to the fermented beverage (stages T0, T3, T6, and PQ) represent 99.58% of the total relative abundance detected (Figure 5, Table 3). All analyzed samples were compared using the Bray–Curtis beta diversity method, and for bacteria, we observed a clear separation between AM and metzal samples from all fermentation stages (Figure 5B). Regarding fungi, AM and metzal samples were partially associated, while all the fermentation stages were found to be closely associated (Figure 5B).

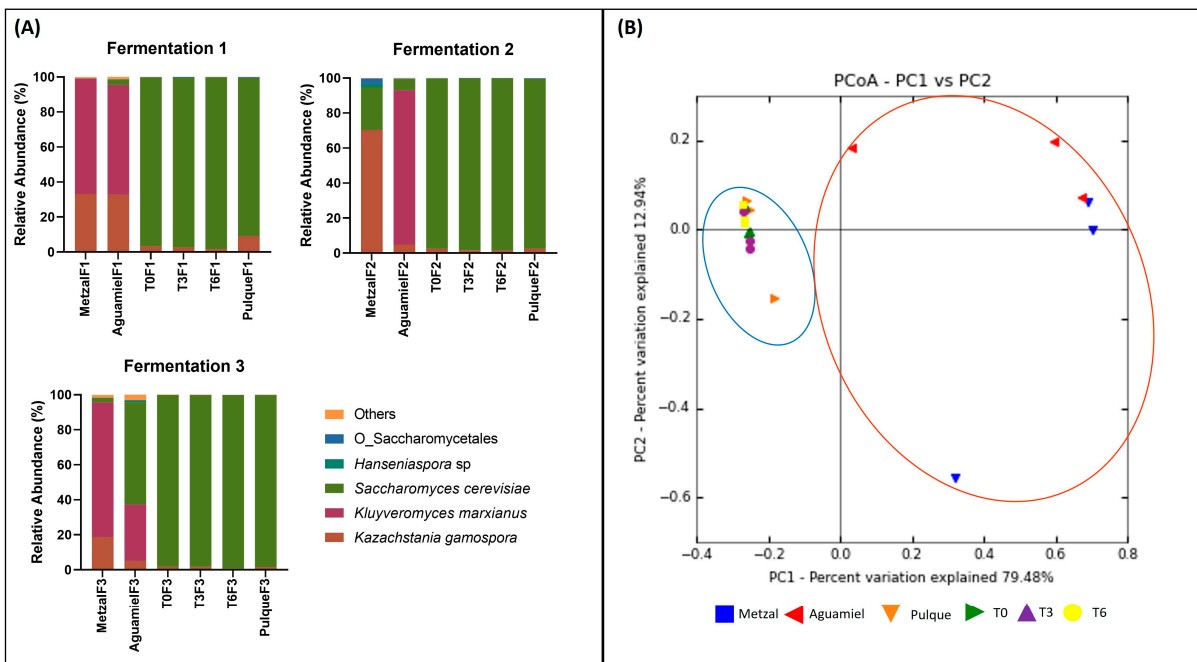

**Figure 5.** Fungi diversity during pulque fermentation. (**A**) The relative abundance profile of fungi is shown for each of the three fermentations using UNITE 2021 database. (**B**) Fungi PCoA plots based on the Bray–Curtis analysis. The blue circle represents an association between the samples. The graphs represent the microorganisms with more than 0.01% of relative abundance. The complete relationship of those microorganisms included in the group Others is shown in the Supplementary Files (Table S2).

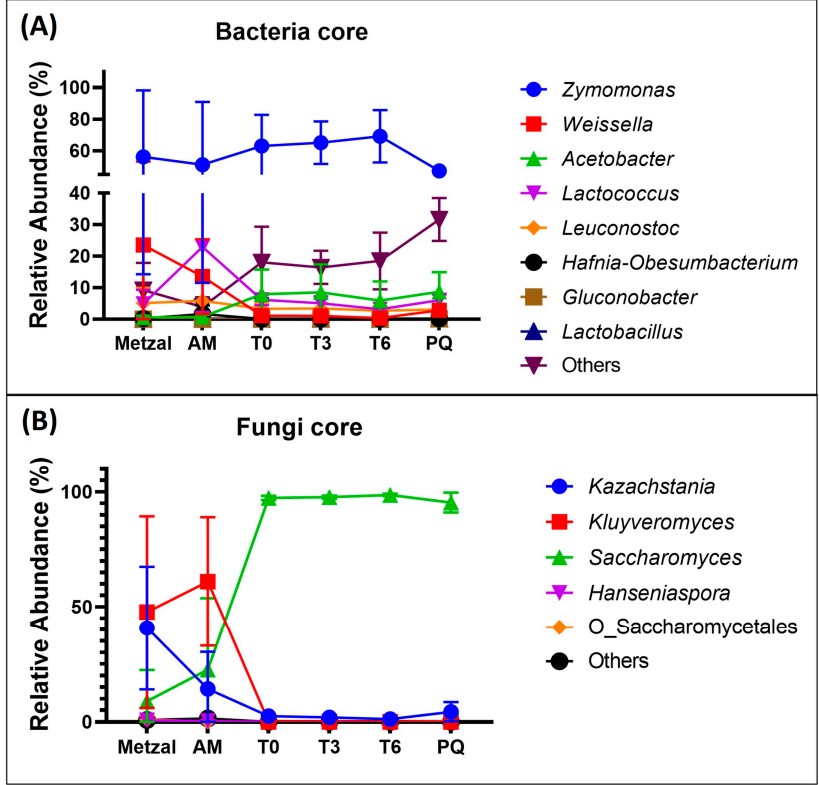

**Figure 6.** Microbial dynamics of bacterial and fungal core in pulque. (**A**) Bacteria. (**B**) Fungi. The graphics represent the average relative abundance of the three analyzed fermentations.

The bacterial and fungal diversity associated with the plant tissue (metzal) of the walls of the cajete and in the sap that accumulated in the cavity after the walls are scraped during the daily process of extraction of AM could be considered as plant-associated diversity, which includes the OTUs considered as the bacterial and fungal core proposed for PQ. Several OTUs included in the Others group were absent in metzal but detected in AM and during all the fermentation stages. These microorganisms include *Lactobacillus acetotolerans*, *L. brevis*, *L. camelliae*, and *L. hilgardii*. In contrast, other OTUs were detected during all the fermentation stages (T0, T3, T6, and overnight fermented PQ), including *Lactobacillus senioris*, *L. similis*, an uncultured *Lactobacillus*, and two OTUs identified as *Acetobacter* (Supplementary Table S1). These findings indicate that these minor abundant bacteria could be incorporated into the sap from sources other than the metzal and incorporated into the fermentation as part of the seed used to start the fermentation.

For fungi, among the 13 OTUs detected in the metzal, five of them are part of the core of the beverage, while the remaining eight (included in the Others group) were only detected in the metzal and AM: *Aurebasidium_pullulans*, *Penicillium* sp., *Turolospora delbruekii*, *Rectipilus* sp., *Malassezia globosa*, *M. restricta*, and the OTUs O_Hypocreales and O_Agaricales. The remaining OTUs identified (five) were only detected in the AM: *Botrytis_caroliniana*, *Erytrobasidium hasegawianum*, *Naganishia albida*, and the OTUs p-Ascomycota and k-Fungi, also suggesting that these minor fungi could be incorporated into the sap from sources other than the *metzal*.

## 4. Discussion

The analysis of the microbial diversity responsible for the fermentation of the traditional Mexican pulque beverage has been studied by diverse authors [9–12,23–27]. Recent metagenomics applications reported the massive sequencing of the entire metagenome of AM and four PQ fermentation stages [11] and the massive sequencing of the 16S rRNA gene V3–V4 fragment as the ribosomal ITSR1 [12]. These reports provided a complete microbiota profile of the beverage and relevant information on the metabolic and functional profiles present in the sap (AM), the substrate for pulque fermentation, as in the different stages of the fermentation analyzed, but did not provide evidence of the possible contribution of the plant's associated microorganisms, particularly those associated with the plant tissue of the walls (metzal) of the cavity or cajete where aguamiel accumulates in producing plants. To our understanding, in this contribution, we analyzed the microbial diversity associated with metzal through massive sequencing of the 16S rRNA gene V3–V4 fragment and ITS region for the first time. We determined which microorganisms, both bacteria and yeasts, present in this tissue are present in AM and maintained during the fermentation process, and their relationship with the physiochemical profile of AM and four PQ fermentation stages.

Bacterial and fungal diversity analysis showed that eight bacterial OTUs identified as *Lactobacillus*, *Leuconostoc*, *Weisella*, *Lactococcus*, *Acetobacter*, *Gluconobacter*, *Zymomonas*, and *Obesumbacterium*, and five fungal OTUs *Kazachstania*, *Kluyveromyces*, *Saccharomyces*, *Hanseniaspora*, and the OTU O_Saccharomycetales were present in metzal and AM, as well as in all the fermentation stages analyzed. These microorganisms could be considered the microbial core for pulque fermentation because they were detected in metzal, AM, and in all the analyzed fermentation stages. Some microorganisms detected in the proposed core were previously reported as maguey endophytes. It is possible to propose that some of them come from the plant, as previous reports support this hypothesis; *L. mesenteroides* was reported as a cultivable endophytic bacteria isolated from the pine of *A. tequilana* [15]. The analysis of the bacterial diversity through 16S rRNA gene V3–V4 fragment amplicon sequencing in the freshly accumulated aguamiel (first 15 mL) after scraping the cajete walls demonstrated the presence of *Leuconostoc* (46.08%) and *Zymomonas* (35.98%) as the most abundant OTUS, comprising 82% of the microbial diversity in this fresh sap [13].

The proposed core comprised up to ~84% of the total bacterial diversity and up to ~99.6% of the fungal diversity detected in the pulque produced from three different plants of *A. salmiana* from the locality of Huitzilac, Morelos, Mexico. The bacterial core found in metzal

comprises 90.75% of the total bacterial diversity for this sample. In AM, the bacterial core comprises 96.10%, and during the 6 h of fermentation (stages T0, T3, and T6), decreased to 81.93%, 83.55%, and 83.70%, respectively, of the total bacterial diversity for these samples (Supplementary Table S4). Finally, the bacterial core in pulque decreased to 68.33% of the total bacterial diversity detected. These results showed the constant composition of the bacterial core from metzal to overnight fermented PQ with variations in the percentage of the specific abundance of some bacterial members of the core during the fermentation process.

The bacterial diversity detected at the genus level agrees with the previous detection through a metagenomic approach in AM and during the fermentation stages in samples provided by the same pulque producer [10,11], except for that fact that our analysis did not detect *Acinetobacter*, reported as the main bacteria present in AM [10,11,24]. Regarding fungi, our results indicated a greater yeast diversity than previously detected [11].

The proposed microbial core dynamics during fermentation indicate that some play a relevant role. *Zymomonas* was the most abundant bacteria in metzal, AM, T0, T3, and T6, with a relative abundance of up to 50%, but decreasing to ~47% in overnight fermented PQ. *Lactococcus* was the second most abundant bacteria OTU detected; it was present in a higher proportion in AM, but its relative abundance decreased after inoculation (T0) with seed (6.1 to 3.1%), and T0 and PQ had similar relative abundance (~6%). *Acetobacter* also increased during fermentation compared to the relative abundance observed in metzal and AM. *Weisella* was detected in a relative abundance above 20% in metzal and above 10% in AM, but it decreased significantly during fermentation. All other OTUs considered members of the bacterial core (*Gluconobacter*, *Leuconostoc*, *Lactobacillus*, and *Obesumbacterium* maintained a relatively constant abundance (below 10%) in all the analyzed samples. Regarding fungi, *Kluyveromyces* and *Kazachastania* were the most abundant yeasts in metzal (Figure 3B). While *Kluyveromyces* increased its abundance in AM and decreased significantly during the fermentation stages, *Kazachastania* decreased in AM and during the fermentation. Remarkably, *Saccharomyces* was present with an abundance below 9% in metzal, increased up to ~22% in AM, and became the most abundant yeast during the fermentation stages.

The AM and the fermentation stages showed an average ash value of 250 mg/100 mL, but with a slight decrease as fermentation developed. Ashes are associated with the amount of inorganic matter, including the minerals in the samples, biomass, and total solids. As fermentation developed, average total solids and biomass decreased despite the relatively constant abundance of the dominant OTUs *Zymomonas*, *Lactococcus*, and *Saccharomyces*. The decrease in biomass as fermentation progressed may be due to the disappearance of residual sugars, FOS, and polymers such as dextran and levan produced during fermentation, although this hypothesis needs to be verified. As an indicator of the metabolic activity in metzal, AM, and during the fermentation stages, we assessed the concentration and consumption of sucrose, glucose, and fructose and the production of the fermentation products ethanol, lactate, acetate, and succinate. Sucrose was practically absent in metzal but showed the highest average concentration in AM (56.6 g/L), and decreased to <1 g/L in T6 and fermented PQ. The detection of fermentation products (ethanol, lactic, acetic, and succinic acids) in metzal was unexpected. However, it could be explained by the metabolic activity of those microorganisms associated with the tissue, since metzal is a hollow tissue and the sap accumulates in the cajete for 10–12 h before being collected, and the walls are scraped again. It is possible that the tissue "traps" the metabolites produced by the associated microorganisms, preventing their diffusion into the aguamiel.

The consumption of sucrose was associated with the production of ethanol, the most abundant fermentative product during PQ fermentation, produced mainly by *Z. mobilis* and *S. cerevisiae*. Both microorganisms transport and hydrolyze sucrose; glucose and fructose moieties enter the Entner–Duodoroff pathway in *Z. mobilis* [28–31] and the Emden–Meyerhoff–Parnas pathway in *S. cerevisiae*, channeling the carbon flux to the alcoholic fermentation to yield ethanol in both microorganisms [32,33]. Although both these ethanologenic microorganisms were reported previously in the metagenomic analysis of AM and during the fermentation

stages for PQ production in samples from the same producing locality (Huitzilac, Morelos) and were correlated with the consumption of fructose and ethanol production, *Z. mobilis* was considered as the main ethanologenic microorganism over *S. cerevisiae* [11]. *S. cerevisiae* is also reported as a succinic acid producer by a limited operation of the tricarboxylic acid cycle, whereas glycerol (also detected in lower concentration in all the analyzed samples) is proposed as a product in redox-balancing reactions or the physiological response to osmotic stress [32]. Although glycerol was detected in lower concentrations, it has been reported to contribute to the desired viscosity of fermented beverages [32].

Regarding the possible relevance of other microorganisms considered part of the microbial core for PQ fermentation, *Leuconostoc* has been widely reported in AM and PQ samples from different geographical origins [9–13,23,24,34] and transports sucrose, which is intracellularly hydrolyzed to glucose and fructose. *Leuconostoc* catabolizes the glucose moiety through the transketolase and heterofermentative pathways to produce ethanol and lactic and acetic acids. *Lactobacillus* and *Lactococcus* transport and catabolize glucose and fructose moieties and channel carbon to the homo- and heterofermentative pathways, contributing to the production of lactic acid, ethanol, and acetic acid [35]. *Gluconobacter* and *Acetobacter* were also previously reported in AM and PQ [10–13], and are considered acetate producers. Acetic acid bacteria produce acetic acid from ethanol oxidation [36,37]. *Gluconobacter* and *Acetobacter* are reported to grow in 1% glucose to produce gluconic acid and keto-gluconates. *G. oxidans* is reported to produce vitamin C, and the genus *Gluconobacter* produces levan from sucrose through an extracellular levansucrase secreted by this microorganism [36,38].

*K. marxianus* was detected as the most abundant yeast in metzal and AM, with relevant metabolic and physiological traits. This yeast was also reported previously in AM and PQ [11,12,39,40]. Like *S. cerevisiae*, *K. marxianus* also produces ethanol and glycerol from sucrose, glucose, or fructose [41,42]. and is anenzymes producer. Among them, inulinases are considered relevant enzymes for FOS production [41].

Fructose and glucose moieties were detected in metzal (an average total = 37.68 g/L for both sugars) despite the low sucrose concentration (average = 0.3 g/L) detected in this sample. As the content of fructans reported in *A. salmiana* was 69% dry basis [42], the detected fructose in metzal and AM may result from the hydrolysis of the inulin in the plant before or during the sap accumulation in the cajete. Alternatively, it may be due to the inversion of sucrose in accumulated AM in the cajete by plant endogenous or microbial enzymes [13,14,40]. As *K. marxianus*, an inulinase-producing microorganism, was detected as the most abundant yeast in metzal and AM, it is a possible candidate to be considered responsible for the hydrolysis of inulin present in the maguey's sap [43].

The average glucose concentration increased slightly from AM to T3 but decreased in T6 and fermented PQ. The fructose concentration increased from AM to T3, with a further decrement in T6 and fermented PQ (Figure 2D). The average amount of sucrose was higher in the AM samples (56.62 g/L), and the total amount (g/L) of glucose plus fructose was 36.31 g/L. Nevertheless, the average sucrose concentration decreased from AM to T3 (until 3.71 g/L), associated with the increase in the average amount of glucose plus fructose during all the fermentation stages (Figure 2D). This behavior suggests that the residual fructose detected during the fermentation stages comes from the hydrolysis of sucrose or possibly from the hydrolysis of the residual inulin from the plant [44], or because of the hydrolysis of several FOS (e.g., nystose ($GF_3$), inulotriose ($F_3$), fructosyl-nystose ($GF_4$), inulopentose ($F_5$)), detected in metzal but absent in AM and during the fermentation stages (Figure 3).

Additional to the possible hydrolysis of FOS detected in metzal, two non-identified peaks, 22 and 23 (between Orafti standards $F_4$ and $GF_5$), were detected in overnight accumulated AM and all the fermentation stages T0–T6, but not in metzal (peak 22). In contrast, peak 23 was detected only in the fermentation times (T0–T6 and fermented PQ). Unidentified FOS corresponding to peaks 22 and 23 could be explained as the result of its biosynthesis from sucrose by a microbial inulosucrase or a fructosyltransferase, such as those reported for *L. citreum* strains [44,45].

Our previous results showed the presence of glucose, fructose, and sucrose in overnight accumulated AM [10,11]. The previous report on the analysis of the sugar content in fresh AM, e.g., the sap accumulated in the cavity in the first 15–20 mL immediately after scraping, showed the presence of sucrose but not glucose or fructose [13], suggesting that sucrose is degraded by microbial activity during the accumulation of the sap, resulting in the accumulation of sucrose moieties. Additionally, differences in FOS profiling in metzal compared to that observed in overnight collected AM, associated with microbial activity, explain the differences in sugar profiles in both samples.

Exopolysaccharide (EPS) production from sucrose, particularly dextran and levan, and their impact on the viscosity of the fermented beverage is a characteristic sensorial trait of pulque [1,2,4,23]. The production of these EPS by microorganisms such as *Leuconostoc mesenteroides* (dextran production), *L. pseudomesenteroides* (dextran production), *L. citreum*, and *L. kimchi* (dextran and levan production) [23,34,46] has been widely reported, as has the production of levan by *Zymomonas mobilis* and *Gluconobacter* [10,28,38].

## 5. Conclusions

In this contribution, we report the presence of a microbial core in three independent pulque fermentations started with aguamiel collected from three *Agave salmiana*-producing plants from the locality of Huitzilac, Morelos state, Mexico. This microbial core comprises up to 84% of the total bacterial diversity and 99% of fungal diversity detected. The analysis of the microbial diversity associated with the walls of the cajete or cavity where aguamiel accumulates indicated that the bacteria and yeast content considered the microbial core for PQ fermentation is associated with this plant tissue and remains in aguamiel during the fermentation. The dynamics of the microbial core during pulque fermentation were associated with the biochemical profile during the process and, remarkably, with the FOS profile. The proposed microbial core could be considered a microbial signature for pulque fermentation.

Further efforts to determine the precise source of those core microorganisms detected in metzal and AM (e.g., plant endophytic or epiphytic source) and the metabolic relevance of those minor microorganisms such as *Obesumbacterium* or those observed in a high abundance in AM but decreasing during the fermentation stages, such as *Weisella* and the yeasts *Kluyveromyces* or *Kazachstania*, are required. Analysis of the proposed microbial core variations associated with other *Agave* species cultivated for pulque production, their different geographical origins, and their seasonal stability is necessary. A microbial core could define the basis for formulating a mixed inoculum for controlled beverage production, maintaining the characteristic sensorial traits.

**Supplementary Materials:** The following supporting information can be downloaded at: https://www.mdpi.com/article/10.3390/fermentation9040342/s1, Supplementary File S1: Biomass quantification by dry weight; Figure S1: Fructooligosaccharide identification per fermentation; Table S1: Bacteria taxonomic identification; Table S2: Fungal taxonomic identification; Table S3: Bacteria taxonomic identification per sample and genus; Table S4: Percentage of core per sample and per genus.

**Author Contributions:** Conceptualization, F.A.-M., F.B. and A.E.; methodology, F.A.-M., G.H.-C. and M.E.R.-A.; formal analysis, F.A.-M., F.B. and A.E.; writing—original draft preparation, writing—review and editing, F.A.-M., F.B. and A.E.; funding acquisition, A.E. All authors have read and agreed to the published version of the manuscript.

**Funding:** This work was supported by projects PAPIIT IN211420 and IN227023 (DGAPA, UNAM).

**Institutional Review Board Statement:** Not applicable.

**Informed Consent Statement:** Not applicable.

**Data Availability Statement:** The raw sequence data generated in this project are available in the National Center for Biotechnology Information Portal as Bioproject PRJNA921737.

**Conflicts of Interest:** The authors declare that the research was conducted in the absence of any commercial or financial relationships that could be construed as a potential conflict of interest.

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
