# Peer review of "Analysis of the Microbial Diversity and Population Dynamics during the Pulque Fermentation Process"

_fermentation, doi:10.3390/fermentation9040342_

Round 1

Reviewer 1 Report

The study is presented at a good level: clear and accurate graphs, charts, all the results are announced in the appropriate section. A lot of work has been done. But, on the whole, the work resembles a training task. The research examines the microbial composition associated with the tissue of the plant walls of the cavity (metzal), where the sap accumulates in producing plants for its daily extraction, in fresh sap (aguamiel), and during four fermentation stages for pulque production. The main thesis of the work is that the microbial diversity of metzal is investigated for the first time and that these microorganisms (bacteria and fungi) form the fermentation process of the beverage. The fact that microorganisms from the sap or plant tissues of the cavity start the fermentation may be true, but the evidences, presented in this study, raise many questions.

1.      A very important and delicate point - the sampling - was made by a local producer of this beverage, not a microbiologist scientist. The article does not reflect:

- how sterile was the sampling (except the sterile bags)?

- how quickly were the samples handed over to the lab?

- could they have started fermenting on their own before the laboratory analysis? and according to the high concentrations of organic acids and ethanol in the metzal, spontaneous fermentation could have begun there.

- how long was the sap in this cavity before sampling?

- how was the metzal separated from the sap? In order to clearly separate plant wall tissues with all the potential phyllosphere and endosphere, it is necessary to be very careful about the sterility of sampling, hands, tools, to conduct a thorough washing of the outer walls from the sap, if it is possible to do it at all.

- how was the “resultant extract” made? could vegetal debris contain the endosymbiotic bacteria?

- the age of the plant could be important. Microbial diversity inhabited their tissues and sap could be different depend on the age.

2.  Only three plants were sampled for analysis. This is not enough considering that one of the three replicates differed significantly in the ratio of core bacteria from the others (Fig. 4a).

3.  It is quite important to know the microbial composition of the primary pulque (“seed” or “starter”), which was used for the laboratory fermentation, moreover it was added more than a half of the volume (“2:3, AM:PQ”). In order to say who exactly influences on the fermentation process you need to know all “zero” points.

4.  In the discussion, the authors repeated their results and collected the known facts about microbes and the processes they conduct, but it all looked raw together. I missed the review on the information about the phyllosphere of agaves (there are such works, for example doi: 10.3389/fmicb.2019.03044), and what was found there? are symbionts inside or outside? has it been shown before that there is so much alcohol and lactate? etc.

Altogether, since there are enough studies about the microbial diversity of AM and different fermentation stages of pulque production (for example very similar works, such as doi: 10.1016/j.micres.2020.126593, 10.1038/s41598-020-71864-4), this research, unfortunately, does not contain enough novelty. Coupled with the fact that the sampling and methodological description of sample preparation is not authoritative, as well as the lack of microbial composition of the starter beverage, all together do not provide a basis for the publication of the results.   

Author Response

The study is presented at a good level: clear and accurate graphs, charts, all the results are announced in the appropriate section. A lot of work has been done. But, on the whole, the work resembles a training task. The research examines the microbial composition associated with the tissue of the plant walls of the cavity (metzal), where the sap accumulates in producing plants for its daily extraction, in fresh sap (aguamiel), and during four fermentation stages for pulque production. The main thesis of the work is that the microbial diversity of metzal is investigated for the first time and that these microorganisms (bacteria and fungi) form the fermentation process of the beverage. The fact that microorganisms from the sap or plant tissues of the cavity start the fermentation may be true, but the evidences, presented in this study, raise many questions.

  1. A very important and delicate point - the sampling - was made by a local producer of this beverage, not a microbiologist scientist. The article does not reflect:

- how sterile was the sampling (except the sterile bags)?

Dear Reviewer,

Thanks for all the concerns about the sampling procedure. The sampling of the metzal and the overnight accumulated aguamiel was made at daybreak, during the daily extraction procedure of the sap made by the local pulque producer, and performed as the traditional process. In our previous papers on the analysis of the microbial diversity in aguamiel and pulque, we reported the same sampling strategy (doi:10.1016/j.ijfoodmicro.2008.03.003, https://doi.org/10.1038/s41598-020-71864-4). In the new version of the manuscript, we described in detail the sampling and sample storage conditions during their transport to the laboratory (new lines 99-122).

- how quickly were the samples handed over to the lab?

The samples of aguamiel and metzal were collected quickly. This traditional operation is no longer than 5 minutes when performed by an expert pulque producer. As the aguamiel was collected and the walls of the cavity of the plant sampled were scrapped, the samples were placed immediately in sterile plastic bags, placed in an ice-filled cooler, and transported by car to our lab located 10 km from the sampling zone in the locality of Huitzilac, Morelos State (25 minutes by car). Details of the sampling procedure and storage during transportation to the is described in the new lines 99-122.

- could they have started fermenting on their own before the laboratory analysis? and according to the high concentrations of organic acids and ethanol in the metzal, spontaneous fermentation could have begun there.

Yes. The fermentation starts as the sap accumulates in the maguey's cavity, as it takes an average of 12 hours. Peralta-García et al. 2020 (https://doi.org/10.3389/fnut.2020.566950) reported Leuconostoc and Zymomonas as the most abundant OTUs in the freshly accumulated aguamiel (first  10-15 mL) immediately after scrapping. The abundance of these bacteria corresponds to 82% of the total diversity detected in the fresh sap. These bacteria are also the most abundant in the accumulated aguamiel. Additional evidence of the microbial activity in the freshly collected aguamiel is the higher degree of polymerization of the FOS detected in metzal compared to that in the fresh sap, suggesting that FOS in aguamiel could be the result of the microbial hydrolysis of that in metzal or resulting from the microbial synthesis (https://doi.org/10.3389/fnut.2020.566950).

To our knowledge, there are no previous reports on the metabolic profile in wet metzal samples. Metzal is a spongy tissue no more than 2 mm thick. The detection of fermentation products was unexpected, but it could be explained by the metabolic activity of those microorganisms in the cajete for 10-12 hours before being collected, and the cavity walls scrapped again. It is possible that the tissue traps" the metabolites produced by the associated, preventing their diffusion into the mead. This explanation was included in the new discussion section in lines 488-494.

- how long was the sap in this cavity before sampling?

The sap is extracted twice a day during the aguamiel production lifetime in producing maguey. One at daybreak, extracting the overnight accumulated sap (12 -14 hours), and a second extraction at twilight, collecting the sap accumulated after the daybreak extraction (10-12 hours). The extent of sap fermentation varies based on the average environmental temperature, rain, dry or cold season. As we report in our contribution, we sampled overnight fermented sap. This information was included in the new manuscript in lines 99-104.

- how was the metzal separated from the sap? In order to clearly separate plant wall tissues with all the potential phyllosphere and endosphere, it is necessary to be very careful about the sterility of sampling, hands, tools, to conduct a thorough washing of the outer walls from the sap, if it is possible to do it at all.

Dear Reviewer,

Thanks for the kind explanation of the procedure to separate plant wall tissues. After the accumulated sap is extracted from the plant's cavity (cajete) at daybreak or twilight, the tissue of the walls is scrapped with a metal tool like a spoon known as a scrapper, and the tissue usually is discarded or used as fodder. Our main objective was not the isolation of endophytic microorganisms present in the pine of the plant where the cavity or cajete is made to promote aguamiel accumulation. We investigated those microorganisms associated with the cavity's scraped walls where the sap accumulates. Our results identified the microbial diversity associated with this plant tissue, considered unique for metzal and absent in the other analyzed samples. Some of these microorganisms could come from the plant, but we require additional experimental evidence to support this hypothesis.

Additionally, all the materials used to squeeze the metzal were sterilized and handled in a sterile environment in a laminar flow hood with gloves to avoid contamination of microorganisms not associated with the traditional fermentation.

- how was the “resultant extract” made? could vegetal debris contain the endosymbiotic bacteria?

The procedure for metzal processing for the isolation of associated microorganisms is described in lines 115-122. Some additional information was included to detail the procedure. As described, the plant tissue was squeezed with a lemon metal squeezer, and the resultant extract was differentially centrifuged (1000 rpm) for the first separation of microbial cells (supernatant) from the complex plant tissue matrix. The supernatant containing the microbial fraction was centrifuged again at high speed to sediment microbial cells for further metagenomic DNA extraction. It is possible that some endosymbiotic bacteria could be sedimented with the plant debris. Although we did not determine the efficiency of the procedure for extracting the microbial fraction associated with the plant tissue, it is possible that the sedimented material contains microbial cells.

- the age of the plant could be important. Microbial diversity inhabited their tissues and sap could be different depend on the age.

We agree with the above comments. To our knowledge, there is not a previously published study profiling the microbial diversity (endophytic or epiphytic) in young to mature maguey plants; of course, this could be valuable information. In our work, we studied the microbial diversity associated with new" or recently prepared plants for aguamiel production (between 5-6 six years old) selected based on the traditional knowledge of the pulque producer who facilitated access to this biological material. This information was included in the new manuscript in lines 110-112.

  1. Only three plants were sampled for analysis. This is not enough considering that one of the three replicates differed significantly in the ratio of core bacteria from the others (Fig. 4a).

Dear Reviewer,

Of course, the number of sampled plants could be considered lower. Nevertheless, the number of plants analyzed was the highest reported to date compared to that included in the recent metagenomic studies on the microbiology of aguamiel and pulque. E.g. the report by Rocha-Arriaga et al. 2020 (https://doi.org/10.1016/j.micres.2020.126593) described the analysis of the microbial diversity in a sole sample of aguamiel from a single plant and two fermentation stages obtained from the locality of Tepeapulco, Hidalgo State, by massive sequencing of the V3-V4 regions from the 16S rDNA. Our previous report on the sequencing of the total metagenome of aguamiel and three fermentation stages included the analysis of one sample of aguamiel from a single plant and for four fermentations stages (T0, T3, T6, and overnight fermented pulque) from the locality of Huitzilac, Morelos, State (Chacón-Vargas et al., 2020, https://doi.org/10.1038/s41598-020-71864-4). Our present contribution included sampling three different plants for metzal and aguamiel supply and developing three independent laboratory fermentations. Each one developed with the sap collected from each producing plant. Additionally, three technical replicates from each stage (metzal, aguamiel, T0, T3, T6, and overnight fermented pulque) from each sampled plant were sequenced as described in lines 174-175 (material and methods section) and in lines 291 and 294 (results section).

  1. It is quite important to know the microbial composition of the primary pulque ("seed" or "starter"), which was used for the laboratory fermentation, moreover it was added more than a half of the volume ("2:3, AM:PQ"). In order to say who exactly influences on the fermentation process you need to know all "zero" points.

Dear Reviewer,

The proportion of aguamiel and fermented pulque used to start the fermentation was suggested by a local pulque producer in the locality of Huitzilac.

The sample of overnight fermented pulque collected together with aguamiel and metzal from each plant was used to prepare the AM:PQ mixture to start the 6-hour fermentation in the laboratory. This strategy was reported previously in doi:10.1016/j.ijfoodmicro.2008.03.003, https://doi.org/10.1038/s41598-020-71864-4(indicated in new lines 135-138).

  1. In the discussion, the authors repeated their results and collected the known facts about microbes and the processes they conduct, but it all looked raw together. I missed the review on the information about the phyllosphere of agaves (there are such works, for example doi: 10.3389/fmicb.2019.03044), and what was found there? are symbionts inside or outside? has it been shown before that there is so much alcohol and lactate? etc.

Dear Reviewer,

In the new discussion section, we highlighted the proposed microbial core for pulque fermentation and the relevance of the analysis of the metzal as the possible source of the core. As indicated, the analysis of the phyllosphere and endosymbiotic profiling of agaves for aguamiel could provide relevant information on those microorganisms associated with the plant transferred" to the fermentation process. Based on our previous results, we explained in the discussion section that isolated endophytes identified as Leuconostoc came from the plant to the sap (new lines 447-454). Additionally, the absence of several bacterial and fungal OTUs in metzal and AM but present in all the fermentation stages supports our proposal. However, as is suggested, it is necessary to explore the plant's phyllosphere, particularly during the producing life of the maguey.

Additionally, we discussed the possible differential role of some microorganisms of the core during the fermentation, associating with the metabolic profile, particularly for metzal (new lines 48-494) and, for the first time, with the FOS profile. Finally, we deleted the original lines 439-446 to avoid redundancy with the results sections.

Altogether, since there are enough studies about the microbial diversity of AM and different fermentation stages of pulque production (for example very similar works, such as doi: 10.1016/j.micres.2020.126593, 10.1038/s41598-020-71864-4), this research, unfortunately, does not contain enough novelty. Coupled with the fact that the sampling and methodological description of sample preparation is not authoritative, as well as the lack of microbial composition of the starter beverage, all together do not provide a basis for the publication of the results. 

Dear Reviewer,

Thanks for the above concerns and criticisms. They improved the quality of the new version of the manuscript considerably. We replied to all of them point by point, including all the concerns about the sampling procedure. Additionally, we included a new conclusion section highlighting the importance of this contribution and further steps in the research of the microbial diversity associated with pulque fermentation (new lines 569-588).

We acknowledge all suggestions based on the previous papers on the profiling of endophytic and phyllosphere microorganisms and their relevance in the microbial core for pulque fermentation. Nevertheless, we consider this the main topic of further research, particularly in aguamiel-producing plants.

Reviewer 2 Report

Authors studied the microbiota profile during fermentation of pulque. In general, the work is very interesting, the aim is clear and the experimental design is appropriate. The manuscript is well-written, the results are well-presented and they are totally supported by the discussion. I only have some minor remarks for further improvement of the paper.

-L95. Better “metabarcoding sequencing”.

-L175-178. What about filtering quality of raw sequences (e.g., denoising, chimera checking, etc.)?

-L282. How many reads corresponded to raw data?

-L327. Please revise.

-L390. From my point of view, the term “population” used herein is not correct. Better use “microbial dynamics” or something relevant.

-L400-405. Please revise.

-L407-409. Genera need italics. Please check throughout the manuscript.

-L527. A final paragraph about a) the importance of the present work, b) how these findings should be exploited by the scientific community and c) the next step(s) about pulque fermentation, is strongly recommended.

-L538. The term “microbiome” is not correctly used. Please revise to “microbiota profile” or something relevant, throughout the manuscript.

Author Response

Authors studied the microbiota profile during fermentation of pulque. In general, the work is very interesting, the aim is clear and the experimental design is appropriate. The manuscript is well-written, the results are well-presented and they are totally supported by the discussion. I only have some minor remarks for further improvement of the paper.

Dear Reviewer, thanks for the above comments and remarks on our manuscript. Please find below the point-by-point reply. Changes are written in red in the new version of the manuscript.

-L95. Better “metabarcoding sequencing”.

The suggested change is included in new lines 93.

-L175-178. What about filtering quality of raw sequences (e.g., denoising, chimera checking, etc.)?

Dear Reviewer, we included in the new lines 189-194 the pipeline used for filtering the quality of sequenced data.

-L282. How many reads corresponded to raw data?

Total raw reads for bacteria were 2,668,499, whereas for fungi were 3,107,428. This information was included in the new lines 303 and 350, respectively.

-L327. Please revise.

Dear Reviewer,

Do you mean the Figure 5 legend in the previous line 327? We modified it to improve the description of corresponding data (new lines 343-344). We also modified the Figure 6 legend similarly (new lines 404-405).

-L390. From my point of view, the term “population” used herein is not correct. Better use “microbial dynamics” or something relevant.

Dear Reviewer,

Do you mean the Figure 6 legend in the previous line, 405? We modified it as suggested in the new line 420.

-L400-405. Please revise.

Dear Reviewer,

Do you mean the information in Table 6? We modified it (new lines 416-417).

-L407-409. Genera need italics. Please check throughout the manuscript.

We checked the entire manuscript for the formal writing of all the scientific names.

-L527. A final paragraph about a) the importance of the present work, b) how these findings should be exploited by the scientific community and c) the next step(s) about pulque fermentation, is strongly recommended.

Dear Reviewer,

A new conclusions section including the suggested perspectives is included in the new version in lines 559-588.

-L538. The term “microbiome” is not correctly used. Please revise to “microbiota profile” or something relevant, throughout the manuscript.

Dear Reviewer, the term microbiome" was replaced by microbiota or other relevant terms in the new version of the manuscript (new lines 83 and 435).

Reviewer 3 Report

 The manuscript is very interesting. Some improvements should be done to increase its overall quality.

Pg. 3 - Inside Table 1, the “Humidity” is about air moisture or about samples? This parameter should be well described inside table.

Pg. 10, Inside table there are no p-values below 0.001. So, the information “** show significant difference with p<0.001.” should be deleted.

Author Response

The manuscript is very interesting. Some improvements should be done to increase its overall quality.

Dear Reviewer, thanks for the above comments on our manuscript. Please find below the point-by-point reply to them. Changes are written in red in the new version of the manuscript.

Pg. 3 - Inside Table 1, the “Humidity” is about air moisture or about samples? This parameter should be well described inside table.

The term Humidity" was replaced by Environmental Humidity” in Table 1.

Pg. 10, Inside table there are no p-values below 0.001. So, the information “** show significant difference with p<0.001.” should be deleted.

This suggestion was attended to in the new Table 2.

Round 2

Reviewer 1 Report

In the revised version of the article the authors took into account most of the comments and strengthened the “Materials and Methods” sufficiently, as well as made the Discussion more informative. However, I still have a few important misunderstandings and questions:

·        It is still very important to know the microbial composition (NGS of 16S rRNA genes) of fermented pulque (as seed) taken to laboratory fermentation process (line 136). The main thesis of the work that “the source of the microbial diversity present in AM and responsible for pulque fermentation comes from the plant tissue and aquamiel” could not be proved without all “zero points” of the experiment. For my point of view, if the authors mixed AM with metzal (or took only AM), allowed it to ferment spontaneously (in the same conditions, as the standard pulque technology process) and the microbial communities of the beverage obtained in this way turned out to be the same as in the pulque produced by the farmer from this plant, then such a conclusion could be made.

·        Also it is quite important to estimate the microbial DNA quantities of AM and fermented pulque (PQ) mixed by the authors to start the laboratory fermentation. If the quantity of microbial DNA from PQ was significantly greater than that of the AM, most likely, the PQ community would prevail in the fermentation process itself, but not the bacteria/fungi of AM or plant tissues. And this seems to explain the same profiles for T0F and Pulque (both bacteria and fungi NGS) (Fig. 4a). The measurement of the biomass and ash (paragraph 2.3 of the Materials and Methods) could be the answer for this misunderstanding, but it was impossible to follow the link (lines 147-148) to estimate what exactly the authors have measured, and Supplementary file did not contain the explanatory information. Unfortunately. There is also a question (I did not pay attention when I read the article for the first time), why the biomass drops during fermentation, what is it related to?

·        It's very difficult for me to understand the boundary between a sap and a metzal when you're essentially squeezing the same sap out of the cavities of the plant walls.

·        The question of the presence of fermentation products (ethanol, organic acids) in metzal remains unexplained to me. The authors suspected that “the tissue traps the metabolites produced by the associated microorganisms, preventing their diffusion to aguamiel”, however this hypothesis did not explain the high concentration of sucrose in the AM and low in metzal (herewith the ratio and composition of the bacterial communities in AM and metzal are very similar in all experiments).     

Despite a great deal of work, there is a little novelty and incontrovertible evidence in the study.

Author Response

Dear Reviewer 1, please find attached the file with the reply point-by-point to all your latest concerns, comments, and suggestions regarding our contribution.

Round 3

Author Response

Dear Reviewer 1,

Please find the reply to the concerns and suggestions merged during the latest review round in the attached file. 
